# H128N Substitution in the Sa Antigenic Site of HA1 Causes Antigenic Drift Between Eurasian Avian-like H1N1 and 2009 Pandemic H1N1 Influenza Viruses

**DOI:** 10.3390/v17101360

**Published:** 2025-10-12

**Authors:** Fei Meng, Zhang Cheng, Zijian Feng, Yijie Zhang, Yali Zhang, Yanwen Wang, Yujia Zhai, Peichun Kuang, Rui Qu, Yan Chen, Chuanling Qiao, Hualan Chen, Huanliang Yang

**Affiliations:** State Key Laboratory for Animal Disease Control and Prevention, Harbin Veterinary Research Institute, Chinese Academy of Agricultural Sciences, Harbin 150069, China; mengfei@caas.cn (F.M.);

**Keywords:** influenza A virus, antigenic variation, hemagglutinin, amino acid substitution

## Abstract

The antigenic relationship between Eurasian avian-like H1N1 swine influenza viruses (EA H1N1) and human pandemic 2009 H1N1 viruses (2009/H1N1) remains a critical question for influenza surveillance and vaccine efficacy. This study systematically investigated the antigenic differences between strains A/swine/Tianjin/312/2016 (TJ312, EA H1N1) and A/Guangdong-Maonan/SWL1536/2019 (GD1536, 2009/H1N1). Cross-hemagglutination inhibition (HI) assays revealed a significant antigenic disparity, with a 16-fold reduction in heterologous versus homologous HI titers. Comparative sequence analysis identified 22 amino acid differences across the five major antigenic sites (Sa, Sb, Ca1, Ca2, and Cb) of the HA1 subunit. Using reverse genetics, a panel of mutant viruses was generated. This study revealed that a single histidine (H)-to-asparagine (N) substitution at residue 128 (H3 numbering) in the Sa antigenic site acts as a primary determinant of antigenic variation, sufficient to cause a four-fold change in HI titers and a measurable drift in antigenic distance. Structural modeling via AlphaFold3 and PyMOL software suggests that the H128N mutation may alter the local conformation of the antigenic site. It is plausible that H at position 128 could exert electrostatic repulsion with adjacent amino acids, whereas N might facilitate hydrogen bond formation with neighboring residues. These interactions would potentially lead to structural changes in the antigenic site. Our findings confirm that residue 128 is a critical molecular marker for the antigenic differentiation of EA H1N1 and 2009/H1N1 viruses. The study underscores the necessity of monitoring specific HA mutations that could reduce cross-reactivity and provides valuable insights for refining vaccine strain selection and pandemic preparedness strategies.

## 1. Introduction

Influenza A viruses, as RNA viruses characterized by high mutation rates, pose a significant threat to global public health due to their continuous evolution and cross-species transmission capabilities. Since its initial isolation from European swine in 1979, the Eurasian avian-like H1N1 swine influenza virus (EA H1N1) has evolved over decades into a stable and geographically distinct lineage, now circulating widely in pig herds across Eurasia [1,2,3,4]. Recent surveillance data indicate an increasing prevalence of this virus in swine populations, establishing it as a major target for regional influenza control efforts. Notably, prevalent EA H1N1 strains exhibit high binding affinity for human-type α-2,6-linked sialic acid receptors on respiratory epithelial cells, providing a molecular basis for overcoming species barriers and enabling zoonotic transmission [5]. Since 2011, sporadic human infections with EA H1N1 viruses have been reported in countries including China, The Netherlands, and Germany, with some cases documenting direct contact with pigs [6,7,8,9,10,11,12,13,14,15]. These incidents confirm the virus’s capacity for human infection and transmission potential, marking it as a candidate virus with pandemic potential.

To date, the 2009/H1N1 pandemic virus, which emerged in 2009 via the reassortment of swine influenza viruses, has been incorporated into the seasonal human influenza cycle and remains a major focus of annual surveillance efforts. Genetic analyses indicate that the hemagglutinin (HA) gene of the 2009/H1N1 virus originated from classical swine influenza lineages, whereas the HA of EA H1N1 viruses is derived from an avian influenza source [16,17]. Although both are classified as H1N1 subtypes, they belong to distinct evolutionary lineages [18,19]. Current serological surveillance data suggest generally low pre-existing immunity in humans against EA H1N1 viruses [4]. The majority of commonly used seasonal influenza vaccines include an antigen similar to A/California/07/2009 as their 2009/H1N1 component; that said, the extent to which these vaccines offer cross-protection against EA H1N1 viruses is still largely unclear.

Antigenicity is a critical determinant of influenza virus immunogenicity and vaccine matching [20,21]. Differences in antigenicity directly impact viral transmissibility, epidemic dynamics, and the effectiveness of control strategies. Serological assays, such as hemagglutination inhibition (HI) and microneutralization (MN), have demonstrated significant antigenic differences between EA H1N1 and 2009/H1N1 viruses. Some EA H1N1 strains show a greater than four-fold reduction in neutralization titers by sera raised against 2009/H1N1 vaccines, meeting the criterion for significant antigenic variation [22]. However, existing studies have been largely limited to in vitro serological comparisons of a small number of strains. The molecular mechanisms by which amino acid variations in key antigenic sites of HA alter epitope structure and neutralizing antibody binding remain poorly understood. The specific residues driving the antigenic divergence between these two lineages and their evolutionary patterns have not been elucidated. Significant antigenic differences would imply that current 2009/H1N1 vaccines may offer limited protection against EA H1N1 infection. Furthermore, in the context of low population immunity, continued antigenic drift in EA H1N1 viruses could facilitate immune escape, potentially leading to localized outbreaks or larger epidemics.

Therefore, we employed serological assays, reverse genetics, and three-dimensional structural modeling to systematically analyze the antigenic differences between EA H1N1 and human 2009/H1N1 influenza viruses. We aim to clarify the molecular mechanisms underlying their antigenic variation and immune escape characteristics. The findings are intended to inform the development of targeted surveillance strategies, facilitate the selection of candidate vaccine strains, optimize seasonal vaccine design, and mitigate the risk of a potential pandemic.

## 2. Materials and Methods

### 2.1. Cells, Viruses, Sera, and Reagents

Human embryonic kidney (HEK293T, ATCC CRL-3216) cells and Madin-Darby canine kidney (MDCK, ATCC CCL-34) cells, maintained in our laboratory, were used for virus rescue and in vitro culture. Both cell lines were cultured in Dulbecco’s modified Eagle’s medium (DMEM) supplemented with 10% fetal bovine serum (FBS, Sigma-Aldrich, Saint Louis, MO, USA), 100 U/mL penicillin, and 100 μg/mL streptomycin at 37 °C in a 5% CO_2_ atmosphere. The human influenza 2009/H1N1 virus strain A/Guangdong-Maonan/SWL1536/2019 (abbreviated as GD1536) was used as the human reference virus. The EA SIV A/swine/Tianjin/312/2016 (TJ312) served as the representative EA H1N1 virus. Additional strains, A/swine/Guangxi/18/2011 (GX18) and A/swine/Guangdong/104/2013 (GD104), representing EA Genetic group 1 and EA Genetic group 2, respectively, were included for antigenic testing. Virus stocks were propagated in the allantoic cavities of 9- to 11-day-old SPF embryonated eggs incubated at 37 °C for 72 h. Allantoic fluid was harvested, clarified by centrifugation at 3000× *g* for 10 min, and titrated for hemagglutination activity before storage at −70 °C for subsequent serological characterization. Detailed background information and antigenic properties of these strains are described in previous studies [2,4].

Ferret antisera raised against TJ312 and GD1536 were used for HI assays. Receptor-destroying enzyme (RDE; Denka Seiken, Tokyo, Japan) was used to remove non-specific inhibitors from serum samples. Cell transfections were carried out using Lipofectamin^TM^ LTX and Plus Reagent (Thermo Fisher Scientific, Waltham, MA, USA). A reverse genetics system based on the PBD plasmid, constructed as described in reference [23], was employed for virus rescue.

### 2.2. Antigenic Analysis

Six-month-old female Angora ferrets were administered intranasally with 10^6^ EID_50_ of TJ312 or GD1536 influenza viruses after anesthesia. 21 days after the infection, blood was collected from the ferrets under anesthesia, and serum was isolated for subsequent experiments. Two ferrets were infected with each virus. Ferret antisera were treated with RDE, serum was mixed with RDE at a 1:3 (*v*/*v*) ratio, incubated at 37 °C for 18 h, and then inactivated at 56 °C for 30 min. Treated sera were serially diluted twofold (starting dilution 1:20). Then, 25 μL of each serum dilution was mixed with 25 μL of virus suspension containing 4 hemagglutination units (4 HAU) and incubated at room temperature for 30 min. Next, 25 μL of a 1% chicken red blood cell suspension was added, followed by an additional 30 min incubation at room temperature. HI was then assessed. The HI titer was expressed as the reciprocal of the highest serum dilution that completely inhibited hemagglutination. Each serum sample was tested against viruses from different antigenic groups. An HI titer difference of ≥4-fold between two viruses was considered indicative of a significant antigenic difference. Each HI experiment was repeated three times.

The antigenic map is a geometric representation of HI assay data created following the instructions (http://www.antigenic-cartography.org/ accessed on 6 June 2025). In such a map, the relative positions of strains (colored balls) and antisera (uncolored cubes) represent the corresponding HI measurements. Distance in the map represents antigenic distance, and the closer the antigens are to each other in the map, the more similar they are antigenically.

The vertical, horizontal, and depth axes all represent antigenic distance, and because only the relative positions of antigens and antisera can be determined, the orientation of the map within these axes is free (thus, an antigenic map can be rotated in the same way that a geographic map can be rotated). The spacing between grid lines is one unit of antigenic distance corresponding to a 2-fold dilution of antiserum in the HI assay. Two units correspond to a 4-fold dilution.

### 2.3. HA1 Protein Sequence Alignment and Analysis

The HA amino acid sequences of GD1536 and TJ312 were obtained from the Global Initiative on Sharing All Influenza Data (GISAID) database. Sequence alignment was performed using the MegAlign module within the Lasergene Suite software (v7.1.0). The amino acid sequences, deduced from the full-length HA genes of both strains, were aligned using the Clustal W method. The homology of the HA1 region was analyzed, with particular focus on amino acid differences within the five known antigenic sites (Sa, Sb, Ca1, Ca2, Cb).

### 2.4. Rescue and Antigenic Validation of Single-Site Mutant Viruses

Recombinant viruses GD1536 and TJ312 were generated using the PBD plasmid-based reverse genetics system. Briefly, a mixture of eight PBD plasmids (250 ng each) was co-transfected into HEK293T cells using Lipofectamine^TM^ LTX and Plus Reagent. At 48 h post-transfection, cell culture supernatants were harvested and inoculated into the allantoic cavities of SPF embryonated eggs. After incubation at 37 °C for 72 h for virus amplification, rescued viruses were subjected to full-genome sequencing to confirm identity with the parental wild-type virus sequences. Chimeric viruses were constructed: chimeric G-T (containing GD1536 HA1 and TJ312 HA2) and chimeric T-G (containing TJ312 HA1 and GD1536 HA2), to determine whether the antigenic differences mapped to the HA1 region. Using TJ312 as the backbone, 43 mutant viruses were generated via single or combined amino acid mutations (Table 1 and Appendix A). With the GD1536 backbone, reverse substitution was implemented for mutant viruses that were suspected of causing antigenic reversal. The antigenic properties of each mutant virus were determined through HI assays. The antigenic map was created using Antigenic Cartography Software (Racmacs 1.2.9, http://www.antigenic-cartography.org), a tool that automatically calculates the antigenic distance based on the coordinates of viruses and sera.

### 2.5. Structural and Mutational Simulation Analysis of the HA1 128 Amino Acid Position

The three-dimensional structures of the HA proteins of TJ312 and GD1536 were predicted using AlphaFold3 (https://alphafoldserver.com/, DeepMind, London, UK) based on their full-length HA amino acid sequences [2]. Default parameters were applied, incorporating evolutionary constraints generated from multiple sequence alignment (MSA). The neural network model was iteratively optimized, and high-confidence structural models were selected. Structure visualization was performed using PyMOL software (v2.5.2). The structural boundaries between HA1 (amino acids 1–330, H3 numbering) and HA2 were defined. All amino acid variation sites within the HA1 protein of both strains and the five key antigenic sites (Sa, Sb, Ca1, Ca2, Cb) were annotated. Mutant models focusing on the amino acid at position 128 (H3 numbering) were constructed to analyze conformational changes in the HA1 head domain before and after mutation.

## 3. Results

### 3.1. HI Assays Reveal Significant Antigenic Differences Between TJ312 and GD1536

Serological analysis results showed that the HI titer of TJ312 antiserum against the heterologous GD1536 virus was 1:40, while its homologous titer was 1:640, representing a 16-fold difference (Table 2 and Figure 1). Conversely, the HI titer of GD1536 antiserum against TJ312 was 1:80, compared to its homologous titer of 1:1280, a 16-fold difference (Table 2 and Figure 1). According to the standard criterion for cross-HI assays (≥4-fold titer difference), these results confirm a significant antigenic difference between TJ312 and GD1536.

### 3.2. HA1 Protein Sequence Alignment Reveals Multiple Amino Acid Differences in Antigenic Sites

The amino acid sequence homology of the HA1 protein between the two strains was 70.6%. The five antigenic sites (Sa, Sb, Ca1, Ca2, Cb) of H1N1 influenza virus HA1 are known to comprise 13, 12, 11, 8, and 5 amino acid residues, respectively (Figure 2 and Figure 3A). Sequence alignment (Figure 2) and amino acid variation sites on the structure of HA1 protein (Figure 3B) revealed that the number of amino acid differences between GD1536 and TJ312 within these sites was 4 (Sa), 6 (Sb), 6 (Ca1), 4 (Ca2), and 2 (Cb), suggesting that multiple amino acid variations in the HA1 antigenic sites likely form the molecular basis for their antigenic divergence.

### 3.3. The Amino Acid at Position 128 in the HA1 Protein Is a Key Determinant of Antigenic Differences Between TJ312 and GD1536 Strains

With TJ312 serving as the backbone, 43 mutant viruses were generated via single or combined amino acid mutations, and 41 of these mutant viruses were successfully rescued (Table 1). The HI assay results of the two constructed chimeric viruses, chimeric G-T and chimeric T-G, indicated that the amino acids responsible for the antigenic differences between the two viruses are located on the HA1 protein (Table 3). HI testing of 41 mutant viruses generated on the TJ312 backbone identified the TJ312/Mut-16 mutant, which exhibited antigenic reversal. Specifically, its HI titer against TJ312 antiserum was 1:160, a 4-fold decrease compared to the homologous wild-type titer (1:640). Its HI titer against GD1536 antiserum was 1:320, a 4-fold increase compared to the homologous GD1536 titer (1:80) (Table 3). Sequence analysis confirmed that this mutant contained a single H128N amino acid substitution (H to N at position 128, H3 numbering) in the HA1 protein.

A reverse mutant, GD1536HA-N128H, was constructed on the GD1536 backbone. HI assays showed that this mutant’s HI titer against TJ312 antiserum increased from 1:40 (wild-type) to 1:320, while its HI titer against homologous GD1536 antiserum decreased from 1:1280 to 1:320 (Table 3). Antigenic cartography analysis indicated that the antigenicity of GD1536HA-N128H drifted from GD1536 toward TJ312. Similarly, the antigenicity of TJ312HA-H128N drifted from TJ312 toward GD1536 (Figure 4). These results confirm that the H128N substitution in the Sa antigenic site of HA1 causes antigenic drift between EA H1N1 and 2009/H1N1 influenza viruses.

### 3.4. Structural Analysis Reveals the Impact of the 128 Amino Acid Substitution on HA1 Conformation

The HA1 domains of both TJ312 and GD1536 exhibited the typical globular head-stem fold, with the head domain consisting of β-sheets and random coils. The Sa antigenic site (residues 127 to 128) was located at the apex of the head domain and was a primary target for neutralizing antibodies (Figure 3 and Figure 5). Structural prediction using AlphaFold3 and visualization with PyMOL software revealed distinct local conformations surrounding residue 128 in the wild-type HA1 proteins of TJ312 and GD1536, with H present in TJ312 and N in GD1536. Based on the high conservation of proline (P) at position 127 and H at position 129, comparative analysis of wild-type and mutant structures, along with hydrogen bond network assessment, allows us to propose a potential molecular mechanism. In TJ312, the imidazole ring of 128H might carry a weak positive charge at physiological pH, which could potentially induce electrostatic repulsion with the adjacent 129H. For GD1536, it is plausible that the polar side chain of 128N could form a hydrogen bond with 129H, possibly contributing to stabilization of a more compact pocket conformation (Figure 5).

## 4. Discussion

Using serology, reverse genetics, and structural modeling, we define the H128N substitution in HA1 as the key determinant underpinning the major antigenic disparity between the EA H1N1 virus TJ312 and the 2009 pandemic H1N1 virus GD1536. This residue is located within the core region of the Sa antigenic site, and its side chain directly contributes to the conformation of the antibody-binding pocket. It is hypothesized that, with adjacent amino acids remaining unchanged, the single amino acid substitution from N to H induces a structural remodeling of the pocket from a compact to a more open state due to the repulsion between positive charges, thereby establishing it as a critical site governing antigenic variation.

Influenza viruses need to evolve continuously via antigenic shift and antigenic drift if they are to keep circulating among humans, as this allows them to escape herd immunity derived from both natural infection and vaccination [24]. According to our laboratory’s publicly available surveillance data from 2013 to 2019 [2]. The 128H has accounted for a certain proportion (10.6%) of Eurasian avian-like H1N1 viruses. Human-isolated 2009 H1N1 strains, however, still predominantly carry the 128N. To date, there have been over forty reported cases of human infection with Eurasian avian-like swine influenza viruses. Moreover, swine influenza viruses carrying the 128H mutation may potentially evade human immunity, posing a risk of human infection with Eurasian avian-like swine influenza viruses. The antigenicity of influenza virus HA protein is primarily determined by five antigenic sites (Sa, Sb, Ca1, Ca2, Cb) within the HA1 domain. Among these, the Sa site, due to its location at the apex of the HA head and its role as a major target for neutralizing antibodies, is a primary driver of antigenic drift [25,26,27,28]. The Sa antigenic site is a key antigenic site for cross-immune protection against the H1N1 influenza virus, and its conservation determines the cross-immune protection across pandemic viruses [29]. Although this study identifies position 128 as a critical residue, the single amino acid change did not fully recapitulate the wild-type antigenic phenotype of the heterologous lineage. HA1 sequence alignment revealed a total of 22 amino acid differences between TJ312 and GD1536 across the five antigenic sites, suggesting that the antigenic divergence between these distinct lineages likely results from a mechanism dominated by key sites with synergistic augmentation of other sites. This finding underscores the diversity and complexity of the molecular mechanisms underlying antigenic change, which is consistent with the observation in previous studies that multiple mutations at antigenic sites promote antigenic drift through a cumulative effect [30]. It highlights the necessity of considering both variations at core sites and the synergistic effects of surrounding residues when evaluating viral antigenicity, particularly for strains with high HA1 homology, where peripheral sites may serve as crucial complementary factors.

## 5. Conclusions

The findings of this study provide a clear molecular marker for the control of EA H1N1 viruses and offer insights into relevant technical strategies. For viral surveillance, the H128N mutation in the HA protein can serve as a core marker to differentiate the antigenicity of EA H1N1 viruses from that of 2009/H1N1 viruses. Furthermore, this study not only elucidates one of the molecular mechanisms underlying the antigenic differences between EA H1N1 and 2009/H1N1 viruses but also clarifies the multi-site synergistic mechanism governing their antigenic distinction. One study has shown that ferrets vaccinated with a human seasonal influenza vaccine were protected against infection with the antigenically matched 2009/H1N1 virus, but not against infection with the EA H1N1 virus [31]. Together with our previous research on the antigenic variation in H1N1 subtype influenza viruses [32,33], these findings provide insights for the precise development of influenza vaccines and the monitoring of cross-species transmission, thereby holding significant theoretical and practical importance for mitigating the risk of influenza pandemics. As influenza viruses continue to evolve, such structure-based, site-specific analyses will remain critical for anticipating antigenic drift and sustaining an effective public health defense system against emerging H1N1 variants.

## Figures and Tables

**Figure 1 viruses-17-01360-f001:**
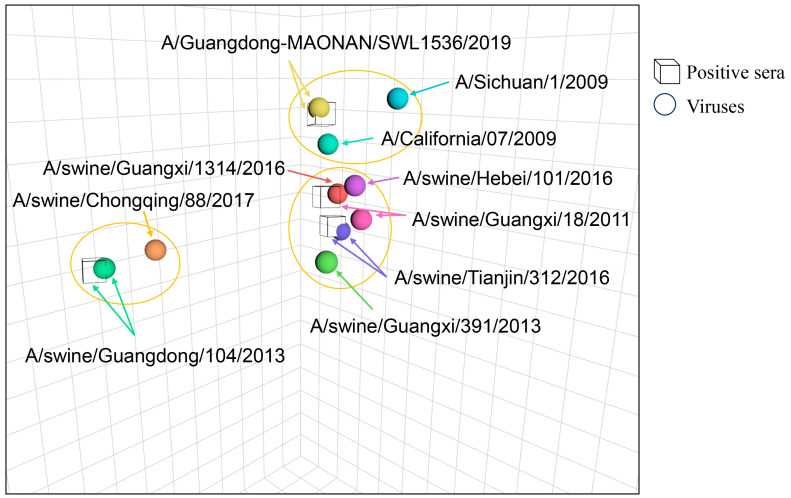
The antigenic map of the H1N1 viruses in this study. The antigenic map was created based on the HI experimental data. Uncolored cubes represent the positive sera raised against the indicated viruses, and colored balls represent different viruses. Each color corresponds to a specific virus strain. The name of each virus strain and antiserum in the chart is individually labeled. The vertical, horizontal, and depth axes all represent antigenic distance. The spacing between grid lines is one unit of antigenic distance corresponding to a 2-fold dilution of antiserum in the HI assay (log_2_HI titers). The viruses within the circles belong to the same antigen group. The antigenic map was created using the antigen software (Racmacs 1.2.9, http://www.antigenic-cartography.org/).

**Figure 2 viruses-17-01360-f002:**
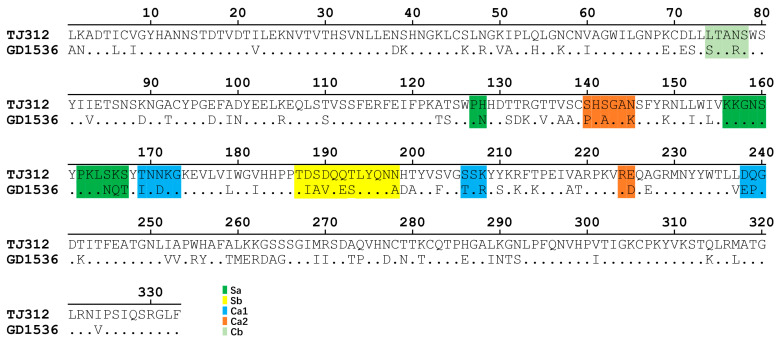
Amino acid sequence alignment of the HA1 proteins of H1N1 influenza virus strains (TJ312 and GD1536). The sequence positions are numbered according to the H3 HA numbering. Sa, Sb, Ca1, Ca2, and Cb, which denote antigenic sites within the HA1 domain, are indicated in green, yellow, blue, orange, and light green, respectively.

**Figure 3 viruses-17-01360-f003:**
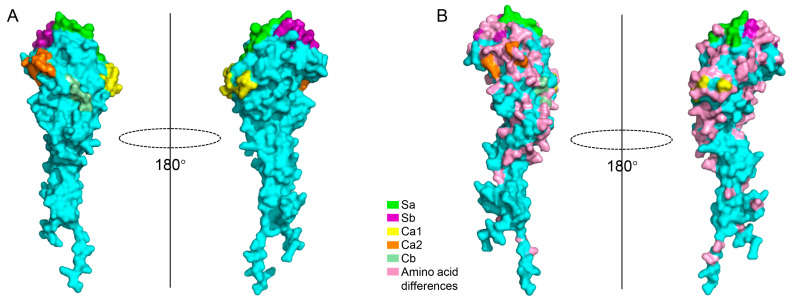
The HA1 structure of TJ312. Alphafold3 was used to predict the HA1 structure of the influenza virus TJ312, and the prediction results were visualized using PyMOL software (v2.5.2). Then, the antigenic regions of HA1 were highlighted with different colors as indicated (**A**), and the different amino acids between TJ312 and GD1536 in the 3D structure were marked in pink (**B**).

**Figure 4 viruses-17-01360-f004:**
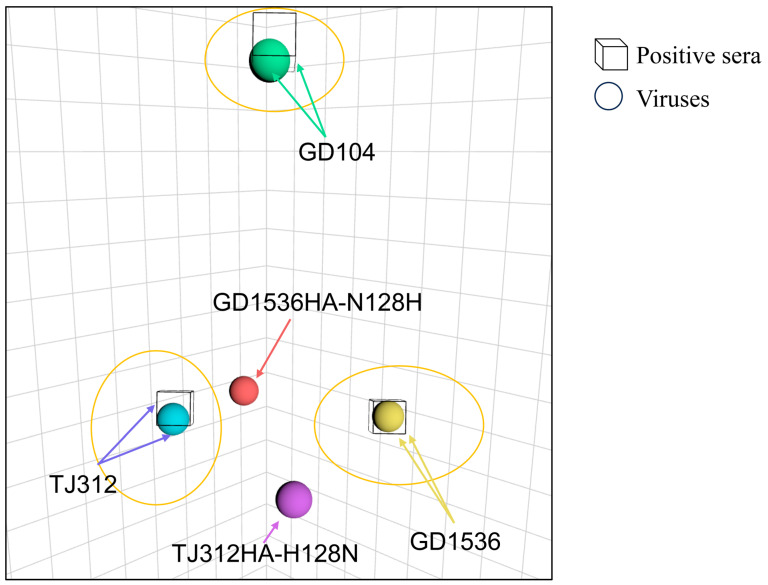
The effect of the 128 N or H in HA on the antigenicity of the H1N1 strain. The antigenic map was created using the same method as in Figure 1, based on the HI experimental data. Each color corresponds to a specific virus. The name of each virus and antiserum in the chart is individually labeled. In this diagram, the three yellow circles represent three antigen groups, and the mutated viruses are marked in red and purple, respectively.

**Figure 5 viruses-17-01360-f005:**
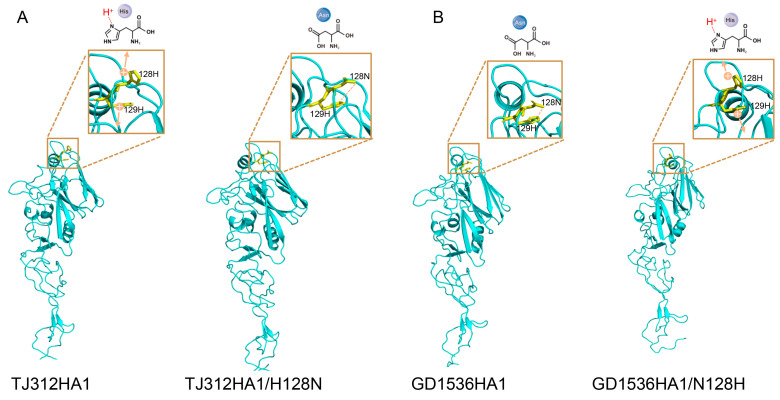
Structural analysis of amino acid site 128 (H3 numbering) of the HA1 head region. The HA1 structures of TJ312 and GD1536 were predicted using Alphafold3, and visualized with the PyMOL software (v2.5.2). The simulated mutations of histidine and aspartic acid were demonstrated in the 3D structure, including TJ312-HA1/H128N (**A**) and GX1536-HA1/N128H (**B**). H: Histidine; N: aspartic acid. Orange and the plus sign denote electrical charge, arrows indicate repulsive interactions, and orange dashed lines symbolize intermolecular hydrogen bonds.

**Table 1 viruses-17-01360-t001:** Viruses with mutations generated in the TJ312 background.

Viruses	Virus Rescue	
	Mutation of HA	Successfully Rescued
TJ312/Mut-1	I6L, V8I	Yes
TJ312/Mut-2	I22V	Yes
TJ312/Mut-3	N38D, S39K	Yes
TJ312/Mut-4	S46K, N48R, K50V, I51A	Yes
TJ312/Mut-5	Q54H	Yes
TJ312/Mut-6	N57K	Yes
TJ312/Mut-7	V60I	Yes
TJ312/Mut-8	K69E, D71E, L72S, L74S, N77R	Yes
TJ312/Mut-9	I83V	Yes
TJ312/Mut-10	K89D	Yes
TJ312/Mut-11	A92T	Yes
TJ312/Mut-12	E97D, A99I, D100N	Yes
TJ312/Mut-13	K105R	Yes
TJ312/Mut-14	T110S	Yes
TJ312/Mut-15	A123T, T124S	Yes
TJ312/Mut-16	H128N	Yes
TJ312/Mut-17	T131S, T132D, R133K	Yes
TJ312/Mut-18	T135V, V137A, S138A, S140P, S142A	No ^a^
TJ312/Mut-19	N145K	Yes
TJ312/Mut-20	R149K	Yes
TJ312/Mut-21	L152I, I154L	Yes
TJ312/Mut-22	S165N, K166Q, S167T, T169I, N171D	No
TJ312/Mut-23	I179L	Yes
TJ312/Mut-24	V182I	Yes
TJ312/Mut-25	D188I, S189A, D190V, Q192E, T193S	Yes
TJ312/Mut-26	N198A, H199D, T200A	Yes
TJ312/Mut-27	S203F	Yes
TJ312/Mut-28	S206T, K208R, Y210S, R212K, T214K	Yes
TJ312/Mut-29	V218A, A219T	Yes
TJ312/Mut-30	E225D, A227E	Yes
TJ312/Mut-31	L237V, D237E, Q239P	Yes
TJ312/Mut-32	T242K	Yes
TJ312/Mut-33	I252V, A253V, W255R, H256Y	Yes
TJ312/Mut-34	A259T, L260M, K261E, K262R, G263D, S264A S265G	Yes
TJ312/Mut-35	M269I, R270I	Yes
TJ312/Mut-36	A273T, Q274P	Yes
TJ312/Mut-37	N277D, T279N, K281T	Yes
TJ312/Mut-38	H286E	Yes
TJ312/Mut-39	L289I, K290N, G291T, N292S	Yes
TJ312/Mut-40	V301I	Yes
TJ312/Mut-41	Q314K	Yes
TJ312/Mut-42	M317L	Yes
TJ312/Mut-43	I324V	Yes

Notes. ^a^ The reverse genetics system was used three times to ensure that the virus cannot be rescued.

**Table 2 viruses-17-01360-t002:** The HI titers of the H1N1 viruses with antisera of different lineage viruses.

Virus	Genetic Group	Ferret Antisera			
		TJ312	GD1536	GD104	GX18
TJ312	EA1	**640** ^a^	80	20	640
GD1536	2009/H1N1	40	**1280**	20	320
GD104	EA2	<20 ^b^	<20	**1280**	20
GX18	EA1	160	160	20	**640**

Notes. ^a^ Homologous titers were marked in bold; ^b^ The lowest detectable level of the serum is 20.

**Table 3 viruses-17-01360-t003:** The HI titers of the chimeric and mutant viruses reacting with the serum of TJ312 and GD1536.

Virus	Ferret Antisera		Virus	Ferret Antisera	
	TJ312	GD1536		TJ312	GD1536
TJ312	**640** ^a^	80	TJ312/Mut-21	640	80
GD1536	40	**1280**	TJ312/Mut-22	-	-
chimeric G-T	40	1280	TJ312/Mut-23	640	40
chimeric T-G	640	80	TJ312/Mut-24	640	80
TJ312/Mut-1	640	80	TJ312/Mut-25	160	20
TJ312/Mut-2	640	40	TJ312/Mut-26	640	80
TJ312/Mut-3	640	80	TJ312/Mut-27	640	160
TJ312/Mut-4	640	80	TJ312/Mut-28	640	80
TJ312/Mut-5	640	80	TJ312/Mut-29	640	40
TJ312/Mut-6	640	80	TJ312/Mut-30	320	80
TJ312/Mut-7	640	80	TJ312/Mut-31	640	80
TJ312/Mut-8	640	80	TJ312/Mut-32	640	80
TJ312/Mut-9	640	80	TJ312/Mut-33	640	80
TJ312/Mut-10	640	80	TJ312/Mut-34	640	40
TJ312/Mut-11	640	80	TJ312/Mut-35	640	80
TJ312/Mut-12	640	80	TJ312/Mut-36	640	80
TJ312/Mut-13	640	80	TJ312/Mut-37	640	40
TJ312/Mut-14	640	80	TJ312/Mut-38	640	80
TJ312/Mut-15	640	80	TJ312/Mut-39	640	80
TJ312/Mut-16	**160**	**320**	TJ312/Mut-40	640	80
TJ312/Mut-17	320	80	TJ312/Mut-41	640	80
TJ312/Mut-18	- ^b^	-	TJ312/Mut-42	640	80
TJ312/Mut-19	640	80	TJ312/Mut-43	640	80
TJ312/Mut-20	640	80	GD1536/N128H	**320**	**320**

Notes. ^a^ Homologous titers were marked in bold; ^b^ The virus was not successfully rescued; G-T is a chimeric virus composed of the HA1 of GD1536 and the HA2 of TJ312, and T-G is a chimeric virus composed of the HA1 of TJ312 and the HA2 of GD1536.

## Data Availability

The data are available upon request.

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
