# Peer review of "H128N Substitution in the Sa Antigenic Site of HA1 Causes Antigenic Drift Between Eurasian Avian-like H1N1 and 2009 Pandemic H1N1 Influenza Viruses"

_viruses, 2025, doi:10.3390/v17101360_

Round 1
Reviewer 1 Report
Comments and Suggestions for Authors
The manuscript presented is a very comprehensive study looking at the antigenic differences between EA H1 and H1pdm09 viruses. The authors did a fantastic job of characterizing the effects of position H128N substitution as an antigenic determinant between these two strains. The study design was very good, and the manuscript represents a large amount of work for this evaluation.
Major flaws- One. The antigenicity differences between these two viruses being centralized to just 128 when so many antigenic sites have substitutions is a bit odd. H1pdm09 like viruses have been known to show narrow antigenic profiles (review https://doi.org/10.1016/j.it.2020.03.005). it was not explained in the methods how many ferrets were used in this study, but could the ferret serum be so narrow as to be mainly directed to that epitope?
Minor issues- 1. Figure 2 is a bit hard to see the changes within in the purple box for the online version of the document. When printed, the dark green, blue and green are nearly impossible to see what the changes are between the two viruses.
2. Was the mutation at 203 (Mut 27) explored? The titer of 160 in the GD1536 is suggestive that position is significant.
3. I think I would minimize the list of mutational primers and enhance the size of the model structures. The small size makes them hard to see and the primers could be shared in a supplemental table.
Reviewer 2 Report
Comments and Suggestions for Authors
This manuscript investigates the antigenic differences between the Eurasian avian-like H1N1 swine influenza virus (EA H1N1) and the human 2009 pandemic H1N1 virus (2009/H1N1). Using cross-hemagglutination inhibition (HI) assays, reverse genetics, and structural modelling, the authors identify a single amino acid substitution (H128N in H3 numbering) in the Sa antigenic site of the HA1 subunit as a key determinant of antigenic variation. The mutation alters local conformation through electrostatic and hydrogen-bond interactions, leading to reduced cross-reactivity. The study underscores the importance of monitoring such mutations for vaccine efficacy and pandemic preparedness. Below are my comments:
- The study focuses heavily on in vitro serological and structural data but provides no in vivo validation (e.g., animal challenge models) to confirm whether the H128N mutation actually affects viral fitness, transmission, or immune escape in a biologically relevant system. How do the authors justify the biological significance of a 4-fold HI titer change in the absence of any in vivo data? Is such a change sufficient to confer immune escape in a natural or vaccinated host?
- The title and abstract claim that H128N "causes antigenic shift," but the data only show a 4-fold change in HI titer—a threshold often associated with antigenic drift, not shift. Antigenic shift typically involves reassortment and major antigenic change. Why do the authors use the term "antigenic shift" instead of "drift", and what evidence supports such a strong claim?
- The structural analysis relies solely on AlphaFold3 predictions without experimental validation (e.g., X-ray crystallography or cryo-EM). AlphaFold models are predictive and may not reflect true conformational dynamics or solvent effects. How confident are the authors that the predicted hydrogen-bonding and electrostatic interactions occur in vivo, especially given the limitations of static modelling?
- The authors generated 43 mutants, but only one (H128N) showed a significant effect. Other mutations may have synergistic or compensatory effects that were not explored. Why did the authors not perform combinatorial mutagenesis to test whether other sites (e.g., in Sb or Ca2) contribute to antigenic variation? Could the lack of effect from other mutations be due to experimental design or assay sensitivity?
- HI assays are semi-quantitative and can be influenced by non-antigenic factors (e.g., receptor binding avidity). No neutralisation assays were performed to confirm the HI findings. Why were microneturalisation (MN) assays not used to validate the HI results, especially given their higher specificity for antigen-antibody interactions?
- The study does not discuss whether H128N is a naturally occurring mutation or how prevalent it is in circulating strains. Without this, its relevance to real-world influenza evolution is unclear. Is H128N a common mutation in surveillance data? Does it correlate with reduced vaccine effectiveness or increased human cases?
- No statistical analysis is provided for HI titers or structural comparisons. Replicates and error bars are absent in figuretitress and tables. How many times were HI assays repeated? What is the inter-assay variability? How were confidence intervals for antigenic distances calculated?
- Figures 1 and 4 (antigenic maps) are poorly explained and lack clarity. The caption for Figure 1 references a previous publication, which is inappropriate for a standalone manuscript.
Table 1: Some mutants were not rescued (e.g., Mut-18, Mut-22), but no explanation is given beyond a note. Was this due to toxicity, instability, or technical failure?
Table 3: The table is poorly formatted and spans multiple pages. It should be restructured for clarity. Homologous titres are bolded, but heterologous changes are not consistently highlighted. This makes it difficult to quickly identify meaningful changes.
Figures 1 and 4: The antigenic maps are not intuitive. The axes are unlabelled, and the grid units are undefined. Also, the use of colours and shapes is confusing. The maps should include a scale bar and an explicit explanation of what each unit represents.
Figure 5: The structural images are low-resolution and lack annotations. Arrows or labels should indicate key residues and interactions. Also, side-by-side wild-type and mutant models would be more informative than separate panels.
Comments on the Quality of English Language
The English could be improved to more clearly express the research.
